# Environmental/Occupational Exposure to Radon and Non-Pulmonary Neoplasm Risk: A Review of Epidemiologic Evidence

**DOI:** 10.3390/ijerph181910466

**Published:** 2021-10-05

**Authors:** Paola Mozzoni, Silvana Pinelli, Massimo Corradi, Silvia Ranzieri, Delia Cavallo, Diana Poli

**Affiliations:** 1Department of Medicine and Surgery, University of Parma, 43126 Parma, Italy; paola.mozzoni@unipr.it (P.M.); silvana.pinelli@unipr.it (S.P.); massimo.corradi@unipr.it (M.C.); silvia.ranzieri@unipr.it (S.R.); 2Centre for Research in Toxicology (CERT), University of Parma, Via A. Gramsci 14, 43126 Parma, Italy; 3INAIL Research, Department of Occupational and Environmental Medicine, Epidemiology and Hygiene, Via Fontana Candida 1, 00078 Monte Porzio Catone, Italy; d.cavallo@inail.it

**Keywords:** radon, epidemiological studies, cancer risk, environmental exposure, occupational exposure

## Abstract

Although Radon (Rn) is a known agent for lung cancer, the link between Rn exposure and other non-pulmonary neoplasms remains unclear. The aim of this review is to investigate the role of Rn in the development of tumors other than lung cancer in both occupational and environmental exposure. Particularly, our attention has been focused on leukemia and tumors related to brain and central nervous system (CNS), skin, stomach, kidney, and breast. The epidemiologic literature has been systematically reviewed focusing on workers, general population, and pediatric population. A weak increase in leukemia risk due to Rn exposure was found, but bias and confounding factors cannot be ruled out. The results of studies conducted on stomach cancer are mixed, although with some prevalence for a positive association with Rn exposure. In the case of brain and CNS cancer and skin cancer, results are inconclusive, while no association was found for breast and kidney cancers. Overall, the available evidence does not support a conclusion that a causal association has been established between Rn exposure and the risk of other non-pulmonary neoplasms mainly due to the limited number and heterogeneity of existing studies. To confirm this result, a statistical analysis should be necessary, even if it is now not applicable for the few studies available.

## 1. Introduction

Radon (Rn), a colorless and odorless radioactive noble gas, originated from the decay of Uranium (U) and Thorium (Th), is found in rocks and soil. Soil is responsible for about 80% of the Rn in atmosphere, water contributes to 19%, and other sources for only 1% [1]. Rn concentration in air depends on the intensity of the source and on dilution factors, both strongly influenced by weather conditions, such as humidity, atmospheric pressure, and wind conditions. Therefore, Rn levels exhibit both daily and seasonal variations, which are often cyclical [2].

Three are the main naturally occurring isotopes of Rn: ^219^Rn (also known as thoron); ^220^Rn (also called actinon); and ^222^Rn, arising from the decay of ^235^U, ^232^Th, and ^238^U, respectively (Figure 1) [3]. The amount of ^219^Rn and ^220^Rn in air pollution is small due to their short half-life (3.96 s, and 55.6 s, respectively) that limits their diffusion in the atmosphere before decay. In addition, the scarcity of ^235^U makes the role of ^219^Rn negligible. Therefore, the lack of knowledge about ^219^Rn and ^220^Rn potential human health effects is due to their unlikely environmental accumulation with consequently reduced human exposure [4], even if their environmental evidence and detrimental effect on human health are described in literature [5]. Most of the radioactivity in the atmosphere attributable to ^222^Rn is due to its longer half-life (3.82 days) that allows its diffusion from environment to dwellings. ^222^Rn decays into more chemically reactive progeny such as ^218^Polonium (Po), ^214^Lead (Pb), ^214^Bismuth (Bi), and ^214^Po (half-life equal to 3,1 min; 26,8 min; 19,9 min; and 1664 µs, respectively), which are able to emit dangerous α and β radiation. Additionally, ^218^Po and ^214^Pb are solids that may spread out in the atmosphere by attaching themselves to air particulate and then settle in soil or water through mechanisms of deposition or the action of rain [6,7].

The concentration of Rn outdoors is typically low and, in any case, does not exceed a few tens of Bq/m^3^ (See Appendix A for Radon measurement units). In fact, Rn escaping from the ground is diluted in a very large volume of air and is rapidly degraded in the atmosphere. On the other hand, in closed places (e.g., homes, offices, schools, etc.) levels vary from a few tens to a few hundred Bq/m^3^, reaching sometimes thousands of Bq/m^3^ [8].

Human exposure occurs primarily through inhalation and ingestion, the latter given by Rn dissolved in groundwater. In the field of occupational exposure, high Rn concentrations can be detected in underground places with poor ventilation and in water treatment plants [9]. The highest concentrations to which workers might be frequently exposed occur in mines; in fact, the first studies related to the effects of Rn exposure have been conducted among miners of the underground mines of uranium [10]. As far as the general population is concerned, Rn exposure is mainly due to its presence in dwellings. Rn levels in building vary regionally and according to season and housing characteristics, with higher concentrations in colder months, in homes with poor ventilation, and on the lower floors of houses [11,12,13,14]. Rn enters buildings through various routes, such as cracks in solid floors, construction joints, cracks in walls above and below ground level, gaps in suspended floors, gaps around service pipes and cavities in the walls [15]. Another cause is the depression created between the various rooms and the ground, induced primarily by the temperature difference between the internal and external environment. The pressure difference determines an upward flow of air from the ground (movement from a high to a low-pressure area), causing the chimney effect [16]. In addition, Rn dissolved in groundwater can be released into indoor air during domestic water use, such as cooking, showering, clothes washing, or water boiling, thereby increasing the total inhalation risk [17,18,19].

Because of its presence in living and working environments and its effects on human health, specific laws and regulations have been produced in many countries. Most European States and many non-European countries recommended reference levels for dwellings and workplaces, and some embraced guidelines for construction techniques and for Rn risk management incorporated in the building codes. In Europe, the International Commission on Radiological Protection (ICRP) and The Council of the European Union (EU) have recommended the Member States to take action against Rn in homes and at workplaces. Several European Directives have been succeeded by lowering the limits required for Rn (e.g., 96/143, 96/29, 2013/59). Consequently, each Member State had to lay down the appropriate provisions, whether by legislation, regulation, or administrative action, to ensure compliance with the basic standards, which have been established. In particular, Council Directive 90/143/Euratom recommendation about indoor Rn in dwellings suggested an average concentration of 200 Bq/m^3^ for new dwellings and 400 Bq/m^3^ in existing dwellings as level for considering remedial action (90/143/Euratom) [20]. The European Council Directive 96/29/Euratom had stipulated that Member States should have required the carrying out of practices for monitoring work activities linked to a significant increase in worker’ exposure (96/29/Euratom) [21] Later, the Council Directive 2013/59/Euratom laying down basic safety standards for protection against the dangers arising from exposure to ionizing radiation repealed Directive 96/29/Euratom establishing 300 Bq/m^3^ as a concentration threshold for both dwellings and workplaces in all EU countries. (2013/59/Euratom) [22]. In the United States, where generally the Rn level is measured in pCi/L (i.e., Ci = Curie that represents the activity of one gram of Rn in radioactive equilibrium; pCi = equivalent to 10^−12^ Curie; 1pCi/l = 37 Bq/m^3^), a Rn level below 2 pCi/L (74 Bq/m^3^) is accepted as normal (Radon Zone 3), while an indoor level between 2 and 4 pCi/L (74–148 Bq/m^3^) is designated as Radon Zone 2 at which the USEPA suggests to perform mitigation. An indoor Rn level above 4 pCi/L (148 Bq/m^3^) (Action Level) is categorized as Radon Zone 1 where mitigation is deemed necessary because an increased risk for lung cancer has been observed at that exposure level [23]. In Western Australia, the management of radioactive materials is governed by the Radiation Safety Act (1975) [24], and the specific provisions relating to the management of naturally occurring radioactive materials in mining operations are included in the Mines Safety and Inspection Act (1994) [25] and Regulations (1995) [26] which specify the same dose limits for exposed workers. Specifically, the effective dose for workers must not exceed 50 mSv (i.e., Sv = Sievert: unit of measurement equal to the absorbed dose of any ionizing radiation having the same biological efficacy as 1 gray of X-rays) in a single year and 100 mSv over a period of five consecutive years. From this scenario, it is clear how Rn mitigation requirements in most states vary substantially. These discrepancies have public health implications. The influence of EU Directives on Rn exposure may have relevance as a model for standardized international regulations. This is why it would be important to harmonize limit value. A Rn assessment is essential because it allows to obtain information on indoor Rn concentration distribution which are useful for decision making (e.g., for establishing reference levels, for individuation of Rn priority areas).

The concern about occupational and environmental exposure arises from the fact that Rn and its short-lived decay products have been classified as a known pulmonary carcinogen in humans by the International Agency of Research on Cancer since 1988 [27].

Epidemiologic studies support a relationship between Rn exposure and cancer risk, in particular for lung cancer [28,29,30]. Although it is conceivable that Rn may have a role in other cancer diseases, the epidemiologic evidence is not so strong such as for lung cancer. One reason is that due to the bio-kinetics of Rn inhalation in the body, the effective radiation doses reaching specific organs is several times lower than that received by the lungs. Furthermore, there is a lower number of scientific publications studying the relationship between Rn and risks other than lung cancer. For the latter, the average of publications on the main databases such as Pubmed [31], Scopus [32], and Web of Sciences [33] is 18.1%, while for the only Rn lung cancer it reaches 81.9%. In addition, most of these studies evaluate the effect of Rn exposure on multiple organs focusing mostly on the lung and making the extrapolation of the results difficult.

The aim of this review is to analyze the epidemiologic evidence in order to investigate the role of Rn in the development of tumors other than lung cancer in both occupational and environmental exposure. Particularly, leukemia and tumors related to brain and central nervous system (CNS), skin, stomach, kidney, and breast have been investigated focusing on workers, general population, and pediatric population.

## 2. Materials and Methods

Published reviews of the epidemiology of Rn that formed the basis of our literature search were by the World Health Organization [28], United States Environmental Protection Agency (USEPA) [34], National Council on Radiation Protection and Measurements (NCRP) [35], National Research Council (NRC) [36], International Agency for Research on Cancer (IARC) [37], and National Institute for Occupational Safety and Health Technical Information Center (NIOSHTIC) [38].

Furthermore, a systematic literature search using PubMed, Scopus, and Web of Science databases was performed with the terms and phrases: “radon” and “cancer”, or “other than lung cancer”, or “brain and Central Nervous System (CNS) cancer”, or “leukemia”, or “skin cancer”, or “stomach cancer”, or “breast cancer”, or “occupational exposure”, or “environmental exposure”. We included original articles, reviews, and meta-analyses published in the English language until 2020. The literature was also searched for regulations related to Rn both in occupational and environmental exposure. We completed the search through consulting manually the references of the papers selected to be full-text read.

## 3. Results

Publications have been analyzed focusing on workers and general and pediatric populations.

### 3.1. Brain and Central Nervous System (CNS) Cancer

Results are summarized in Table 1a–c.

#### 3.1.1. Workers

Almost all studies in working population were performed in miners. In 1995, Tirmarche et al. studied French uranium miners exposed to Rn concentrations range from 500 to 1000 Bq/m^3^ in order to calculate the number of expected deaths due to brain cancer. Combined malignant brain tumor, malignant tumors of other parts of the nervous system, and tumors of unspecified nature of the brain and other parts of the nervous system were investigated. A significant risk of death (*p*-value = 0.03) has been calculated only by excluding the last group (i.e., tumors of unspecified nature of the brain and other parts of the nervous system) [39]. Darby et al. examined mortality from non-lung cancer in an analysis of data from 11 cohorts of underground miners in which Rn-related excess of lung cancer had been established. The study included 64,209 men employed in the mines for 6.4 years on average, receiving average cumulative exposures of 155 working-level months (WLM), and were followed for 16.9 years on average. This study provides considerable evidence that high concentrations of Rn in air do not cause a risk of mortality from cancers other than lung cancer, including brain and CNS cancers [40]. Similarly, in a cohort of iron miners from northern Sweden occupationally exposed to elevated levels of Rn, the mortality was increased for all cancers other than lung cancer, but it was not significantly associated with cumulative exposure to Rn [41]. About ten years later, Vacquier and co-authors in a first study on a cohort of men employed as uranium miners between 1946 and 1990 highlighted a significant excess risk of cancer death for a lung (associated with levels of cumulated radon exposure) and kidney (not associated with radon exposure), but the brain and CNS cancer had not been included [42]. In a later paper, the same authors examined the mortality risks associated with exposure to Rn, external γ- rays, and long-lived radionuclides (LLR) in the French “post-55” sub-cohort, including uranium miners first employed between 1956 and 1990 for whom all three types of exposure were individually assessed. The study highlighted for the first time an increase of mortality for brain and CNS cancer (SMR: 2.00; 95% CI: 1.09–3.35) [45].

On the contrary, the cohort of uranium miners examined from Kreuzer et al., which included 58,987 men employed for at least 6 months from 1946 to 1989 in an uranium mining company in Eastern Germany, highlighted a statistically significant increase in mortality for other than lung cancers (stomach and liver), but not for brain and CNS [43,58]. Finally, in the Colorado Plateau cohort (3358 white miners and 779 miners of another race) a significant risk to develop some cancer types was found but not brain or CNS cancers [44].

#### 3.1.2. General Population

In 1993, Hess et al. demonstrated significant correlation between Rn levels throughout the countries of the state of Maine, USA, and incidence of all cancers, including brain and nervous system cancer [59]. Similarly, twenty years later, a Danish study in a cohort of 57,053 persons observed a statistically significant association between residential Rn and brain cancers. The adjusted incidence rate-ratios (IRR) for primary brain tumor associated with each 100 Bq/m^3^ increment in average residential Rn levels was 1.96 (95% CI: 1.07–3.58). This association was not modified by air pollution [47]. Later, a paper from a Spanish group observed a significant correlation between residential Rn exposure and brain cancer mortality, with a higher correlation for females. These results were reinforced when the analysis was restricted to municipalities with more than five Rn measurements, showing Spearman’s Rho equal to 0.286 (*p*-value < 0.001) and 0.509 (*p*-value < 0.001) for males and females, respectively [48]. Again, another Spanish study highlighted that indoor Rn concentration in Galicia was statistically associated with higher lung, stomach, and brain cancer mortality only among women [49].

In contrast, Monastero et al. reported no relationship between mean Rn levels and CNS cancer incidence in five highly populated and Rn-enriched US states [50], as well as the Turner’s study performed in the American Cancer Society cohort [46].

#### 3.1.3. Pediatric Population

In 1991, the association between groundwater Rn levels and childhood cancer mortality in North Carolina was explored. This study highlighted an increase of the relative risks for several cancers, included brain and CNS tumors [51]. Another study considered children suffering from leukemia and common solid tumors (nephroblastoma, neuroblastoma, rhabdomyosarcoma, and CNS tumors) diagnosed between 1988 and 1993 in Lower Saxony, Germany. Rn measurements were performed for one year in those homes where the children had been living for at least one year, with particular attention posed to those rooms where they had stayed most of the time. The risk estimates were high for solid tumors (OR: 2.61; 95% CI: 0.96–7.13) [52].

In contrast with these results, in a Danish children cohort, the cumulative Rn exposure was not associated with risk for CNS tumor [53], as well as in a Swiss study where a cohort of 997 childhood cancer cases was evaluated. Specifically, compared with children exposed to Rn concentration below the median (<77.7 Bq/m^3^), adjusted hazard ratios for children with exposure ≥ the 90th percentile (≥139.9 Bq/m^3^) were 0.93 (95% CI: 0.74–1.16) for all cancers and 1.05 (95% CI: 0.68–1.61) for CNS tumors [54]. Similarly, Kendal et al. reported how Rn exposure was not significant for brain and CNS childhood cancers in children born and diagnosed with cancer or nonmalignant brain tumor in Great Britain between 1980 and 2006, as recorded on the National Registry of Childhood Tumors [55]. Again, in a Norway cohort of 712,674 children with a total of 864 cancer cases 427 of them related to the CNS, an elevated non-significant risk for cancer was observed [56]. An ecological study related to a cohort of 5471 children with CNS tumors demonstrated that there was no association between Rn exposure and childhood CNS tumors incidence (IRR: 1.07; CI: 0.95–1.20 per 100 Bq/m^3^) [57]. Finally, the results of a review performed on 18 studies (8 on miners, 3 on the general population, and 7 on children) are inconclusive because the available studies are extremely heterogeneous in terms of design and populations [60].

### 3.2. Leukemia

The main results are described in Table 2a (Workers) and Table 2b (General and Pediatric Population).

#### 3.2.1. Workers

A report of Tomasek et al. demonstrated an increased mortality trend from multiple myeloma with cumulative exposure to Rn and an increasing trend of leukemia mortality with long-lasting employment in uranium mines. Mortality from multiple myeloma, although not significantly increased overall, increased with cumulative exposure to Rn. Instead, mortality from leukemia was not increased overall and was not related to cumulative Rn exposure but did increase with increasing duration of employment in the mines [61]. In a pooled statistical analysis combining 11 epidemiological studies on underground uranium miners, Darby et al. established increased leukemia mortality only in the period of less than 10 years after beginning work at mine [40]. In a case-cohort study in Czech uranium miners, exposure to Rn and its progeny was associated with an increased risk of developing leukemia. Particularly, an increased incidence of all types of leukemia, along with chronic lymphocytic leukemia, in relation to cumulative Rn exposure has been observed. The relative risk (RR) comparing high Rn exposure (110 WLM; 80th percentile) to lower Rn exposure (3 WLM; 20th percentile) was equal to 1.75 (95% CI: 1.10–2.78; *p*-value = 0.014) for all leukemia subtypes combined and 1.98 (95% CI: 1.10–3.59; *p*-value = 0.016) for chronic lymphocytic leukemia (CLL) [62].

Kreuzer et al. analyzed a cohort including 58,987 men employed for at least 6 months from 1946 to 1989 at the former Wismut uranium mining company in Eastern Germany. The number of deaths observed for leukemia was close to that expected from national rates. No association between cumulative Rn exposure and leukemia was found, or with chronic lymphatic leukemia (CLL), non-CLL or acute myeloid leukemia (AML) [43].

Zabloska et al. analyzed radiation-related risks of hematologic cancers in the cohort of Eldorado uranium miners and processors first employed in 1932–1980 in relation to cumulative Rn decay products (RDP) exposures and γ-ray doses. The average cumulative RDP exposure was 100.2 working level months (WLM). No statistically significant association between RDP exposure or γ-ray doses, or a combination of both, and mortality or incidence of any hematologic cancer was found [63].

A cohort of 16,434 male underground miners from Czech Republic were exposed to low and moderate levels of Rn gas and other hazards. The SIR was elevated for all leukemias (SIR: 1.51; 95% CI: 1.08–2.07) and for lymphatic and hematopoietic cancers combined (SIR: 1.31; 95% CI: 1.05–1.61) [64].

#### 3.2.2. General Population

With regard to environmental exposure, in 1989 Lucie suggested a relationship between Rn concentration and the incidence of leukemia in England and Wales [73]. Subsequently, significant correlations between Rn concentration and several leukemia subtypes in England and Wales have been observed [74]. Similarly, Eatough et al. observed a significant correlation between standardized registration ratio (SRR) for monocytic leukemia and the Rn concentration by county in England. The authors showed these results had been unlikely produced by regional variations in registration efficiency or by being confounded due to social class or to gamma radiation exposure [65]. In line with these data, Henshaw et al., suggested that for the world average Rn exposure of 50 Bq/m^3^, 13–25% of myeloid leukemia at all ages might be caused by Rn [70].

#### 3.2.3. Pediatric Population

Between 1976 and 1985 in Britain, Thorne and colleagues evaluated the incidence of childhood malignancies compared to postcode sectors with a Rn exposure ≥ 100 Bq/m^3^ with sectors with a Rn exposure < 100 Bq/m^3^. No significant difference in the incidence rate as compared to all cancers and no association between Rn exposure and overall rate of childhood malignancy were found [66]. Steinbuch et al. evaluated the risk factors for childhood acute myeloid leukemia associated with indoor residential Rn level within a larger interview-based case–control study performed over 120 institutions in the USA and Canada. A total of 173 cases and 254 controls were analyzed and no association was observed between Rn exposure and risk of acute myeloid leukemia, with adjusted OR of 1.2 (95% CI 0.7–1.8) for 37–100 Bq/m^3^ and 1.1 (95% CI: 0.6–2.0) for >100 Bq/m^3^ compared with <37 Bq/m^3^ [67]. In another study, including 505 cases and 443 age matched controls, the association between the incidence of acute lymphatic leukemia (ALL) in children under age 15 years and indoor Rn exposure was investigated. Mean radon concentration was lower for case subjects (65.4 Bq/m^3^) than for control subjects (79.1 Bq/m^3^). Therefore, the results from this analytic study provide no evidence for an association between indoor Rn exposure and childhood ALL [68].

Kaletsch et al. conducted a case/control study on 82 cases of childhood leukemia in Lower Saxony. Long term Rn measurements were carried out in dwellings where the children had lived for at least one year. There was no association between higher radon levels and leukemia [OR: 1.30; 95% CI: 0.32–5.33)] [52].

One of the largest of the available case/control studies was conducted in the United Kingdom. The parents of 3838 children with cancer (1461 of which were of acute lymphoblastic leukemia) and of 7629 children without cancer were interviewed. The arithmetic mean Rn concentration measured in the homes was 24.0 Bq/m^3^, with the mean concentration being slightly lower in case homes than in control homes. No evidence to support an association between higher Rn concentrations and risk of any of the childhood cancers was then found [69].

In contrast, the analysis performed by Henshaw et al. for myeloid leukemia in UK children suggested that the 6–12% incidence might be attributed to Rn exposure, the figure becoming 23–43% in Cornwall, where Rn levels are higher [70]. In the same way, for data from throughout Britain, a significant correlation for childhood leukemia with Rn concentration by county was observed [75]. Gilman and Knox conducted a study about 8500 cases of childhood cancers diagnosed up to age 15 and born in Great Britain between 1953 and 1964. There was a significant positive trend of mortality with increasing Rn exposure for leukemias and lymphomas (RR. 1.06, CI 0.99–1.12), just failing to reach statistical significance [71].

In 2000, a Swedish study assessed 53,146 children, and among these, 90 developed hematologic malignancies. Standardized mortality ratios (SMR) for acute lymphatic leukemia among children born in high, normal, and low risk areas were 1.43, 1.17, and 0.25, respectively. The RR for normal risk group and high-risk group as compared with the low-risk group was 4.64 (95% CI: 1.29–28.26) and 5.67 (95% CI: 1.06–42.27). The association between acute lymphatic leukemia and continued residence in normal or high-risk areas showed a similar trend. No association between Rn risk levels and any other malignancy was seen. This study evidenced that children born in and continuously living in areas classified as high and normal risk for background radiation of Rn have a higher incidence of acute lymphatic leukemia [72]. Later in 2012, a meta-analysis of case-control studies was conducted to uncover the influence of Rn exposure on childhood leukemia. The combined OR for calculating the lowest exposure of Rn on the high incidence of childhood leukemia was 1.37 (95% CI: 1.02–1.82), suggesting a weak association. A major source of uncertainty was the Rn dose estimate [76]. However, in a 2016 review about environmental exposure and risk of childhood leukemia, Schüz concluded that radiation is not a contributor to the global childhood leukemia burden [77].

### 3.3. Skin Cancer

As regards Rn exposure and skin cancer, scientific works are mainly focused on the general population (See Table 3).

#### General Population

The American Cancer Prevention Study II (CPS-II), a large prospective study of nearly 1.2 million participants recruited in 1982 by the American Cancer Society, highlighted no association between residential Rn and any other mortality cause beyond lung cancer or chronic obstructive pulmonary disease. With regard to the relationship with skin cancer, the authors calculated HRs of 1.08 (95% CI: 0.88–1.33) and 0.70 (95% CI: 0.42–1.19) per 100 Bq/m^3^ in mean county-level residential Rn for malignant melanoma and non-melanoma skin cancer mortality, respectively [46]. In 1996, Etherington et al., studying the relationship between domestic Rn levels and cancer in southwest England, observed that only non-melanoma skin cancers showed a significant increase in incidence in the high-Rn sectors (≥100 Bq/m^3^) compared to the low-Rn sectors (<60 Bq/m^3^) [78]. Later, still in the southwest of England, the ecological epidemiology data provided no evidence for elevated skin cancer risks at Rn levels <100 Bq/m^3^ [84]. Wheeler et al. found an association between Rn levels in Southwest England and incidence of squamous cell carcinoma (SCC), but not for basal cell carcinoma (BCC) or malignant melanoma [79]. The same authors did not find an association with incidence of non-melanoma skin cancer (SCC and BCC combined) [80]. In a cohort study in Galicia performed on 2294 subjects, no association appeared for several tumors except for melanoma [82]. In a Danish cohort, Bräuner et al. found a statistically significant association between basal cell carcinoma and Rn (IRR: 1.14; 95% CI: 1.03–1.27) but not for squamous cell carcinoma (IRR: 0.90; 95% CI 0.70–1.37) and malignant melanoma (IRR: 1.08; 95% CI: 0.77–1.50) [81]. In contrast, in a study conducted in Switzerland, a significant positive association with malignant melanoma mortality and Rn exposure was observed with an HR of 1.16 (95% CI: 1.04–1.29) per 100 Bq/m^3^ [83]. Finally, Denman et al., considering the Rn levels in abandoned mines and the lung and skin dose received by visitors, highlighted the increased lifetime risk of skin cancer due to a 1 h exposure in each mine. The results indicated that if visitors ensure that their Rn exposure is below the National Radiological Protection Board (NRPB) guideline of 106 Bq/m^3^ h in 1 year, the skin dose and effective dose will both be below the annual limits for UK radiation workers and below the levels at which any acute skin effect will occur [85].

### 3.4. Stomach Cancer

Table 4 summarized the main results related to workers and general population.

#### 3.4.1. Workers

Deaths from several cancer types, including stomach tumor, are elevated among the miners. In a collaborative analysis of data from 11 cohorts of underground miners in which Rn-related excesses of lung cancer had been established, Darby et al. examined the mortality from non-lung cancer. The study included 64,209 men who were employed in the mines for 6.4 years on average. Among 28 individual cancer categories, statistically significant increases in mortality for cancers of the stomach (O/E: 1.33; 95% CI: 1.16–1.52) were observed [40]. Similarly, a cohort of underground miners in the Czech Republic between 1977 and 1992 was selected from the registry Příbram Uranium Industry (UI) employees. There was a 52% increase in deaths from all malignant causes compared with expected rates (SMR: 1.52, 95% CI: 1.44–1.60). In addition to lung cancer, mortality was also higher for other cancer subtypes, particularly stomach cancer (SMR: 1.27; 95% CI: 1.02–1.51), as well as all cancer incidence, stomach cancer included (SIR: 1.37; 95% CI: 1.11–1.63) [64]. In Swedish iron miners occupationally exposed to elevated levels of the radioactive gas Rn, mortality was increased for all cancers other than lung cancer. With regard to stomach cancer, ratio of observed to expected deaths was 1.45, with a 95% C.I. of 1.04–1.98. However, the authors underlined that the increase in stomach cancer mortality might be due to the considerable number of Finns in the workforce, in which stomach cancer rates were considerably higher than in Swedish population. In addition, mortality was not significantly associated with cumulative Rn exposure [41]. Finally, in a cohort of 28,546 workers in Ontario uranium mines, no association was observed between increasing cumulative Rn exposure and other than lung cancer (e.g., stomach, leukemia, kidney, and extrathoracic airways) [86].

#### 3.4.2. General Population

In a retrospective analysis that compared Rn levels for each county in Pennsylvania to the incidence and mortality of gastrointestinal cancer, a positive correlation was found for stomach cancer in females, and the mortality of stomach cancer for male, female, and total population [88]. Messier et al. provided epidemiological evidence of a positive association between groundwater Rn concentration and an increase in the probability of a stomach cancer [87]. Moreover, Lopez-Abente evaluated the cancer mortality in Galicia and residential Rn levels. For the whole study period (1999–2008), mortality data for each of the 313 Galician municipalities were drawn from the records of the National Statistics Institute. Expected cases were computed using Galician mortality rates for 14 types of malignant tumors as reference, with a total of 56,385 deaths due to the tumors analyzed. The effect estimates of indoor Rn (3371 sampling points) were adjusted for several parameters, such as sociodemographic variables, altitude, and arsenic topsoil levels. The results showed a statistical association between twofold increase in indoor Rn and several cancers in women, included stomach cancer (RR: 1.174; 95% CI: 1.022–1325) [49]. In contrast, an epidemiologic study conducted in Finland to evaluate the risk of developing stomach cancer due to the presence of Rn in drinking water showed no association (HR = 0.68, 95% CI: 0.29 –1.59 at 100 Bq/L water), even for Rn concentrations exceeding 300 Bq/L. This study was conducted for the very high concentrations of several naturally occurring radionuclides in the Finland ground. Only subjects using drilled wells as a source of drinking water were included in the study, and 1492 were the stomach cancers diagnosed during 1981–1995 from the Finnish Cancer Registry [89].

### 3.5. Kidney Cancer

Few scientific works studied Rn exposure and kidney cancer and they are mainly focused on workers (see Table 5).

#### Workers and General Population

In an extended follow-up study (1946–1990) performed on 4140 French miners exposed to Rn for at least 1 year with an average cumulative exposure of 36.6 WLM, a significant excess of kidney cancer deaths was observed (SMR: 2.0; 95% CI: 1.22–3.09). However, no association with cumulative Rn exposure was found [42]. Similarly, Navaranjan et al., evaluating an Ontario uranium miner cohort consisting of 28,546 male miners with a mean cumulative Rn exposure of 21.0 WLM, observed an association with Rn exposure only for lung cancer [86]. Similar results were obtained in French (*n* = 3377) and German (*n* = 58,986) cohorts of uranium miners where deaths from kidney cancer were analyzed. In detail, no significant excess of kidney cancer mortality had been observed neither in the French cohort (SMR: 1.49, 95% CI: 0.73–2.67) nor in the German cohort (SMR: 0.91; 95% CI: 0.77–1.06]). Moreover, no significant association between kidney cancer mortality and any type of occupational Rn exposure or kidney equivalent dose had been observed [90]. Finally, in a careful and precise meta-analysis, Chen et al. analyzed several studies carried out in order to verify the relationship between kidney cancer and Rn exposure. Authors concluded how it was not possible to affirm a clear correlation mainly for the lack of homogeneity in the exposure assessment, for the different sample sizes (from 779 to 28,546) and because in many cases kidney cancer mortality rather than incidence was evaluated, thus underestimating the carcinogenic effects of Rn on the kidney. In addition, most of the studies were carried out on mine workers with the possibility of co-exposure to other carcinogenic mineral products such as arsenic, silica dust, and gamma radiation [91].

### 3.6. Breast Cancer

Results, mainly focus on general population, are reported in Table 6.

#### Workers and General Population

Increased breast cancer incidence was observed among former female employees of a Missouri school with elevated Rn levels [93]. In contrast, ecologic studies in the U.S. showed no association between county-level Rn concentrations and breast cancer incidence [95]. Later, a perspective analysis carried out in the American Cancer Society cohort showed no association between Rn exposure and breast cancer-related mortality [46]. Recently, also in the USA, VoPham et al. examined the association between environmental Rn levels and breast cancer incidence in a prospective cohort of non-occupationally exposed of 112,639 females. Increasing Rn exposure was not associated with breast cancer risk overall. However, women in the highest quintile of exposure (≥74.9 Bq/m^3^) showed an elevated risk of ER−/PR- breast cancer compared to women in the lowest quintile (<27.0 Bq/m^3^). No association was observed for ER+/PR+ breast cancer [95]. Finally, breast cancer incidence in Iceland resulted higher among residents of high temperature geothermal areas and higher level of Rn in water compared to residents of non-geothermal areas [93].

## 4. Discussion

The possible etiologic relationship between Rn and tumor sites other than lung, as well as the pathogenic mechanism, are controversial. In this review, we analyzed the epidemiologic evidence in the development of tumors other than lung cancer in both occupational and environmental Rn exposure. Several large-scale studies were launched in an effort to investigate factors possibly affecting dose–response relationship and Rn exposure in other than lung cancer. As far as occupational exposure is concerned, literature data refer almost exclusively to miners, while for environmental exposure, the research work focuses on both inhalation and ingestion exposure due to the presence of Rn in water. For their greater risk of developing cancer diseases because of radiation exposure compared to adults, the outcomes of studies related to children were also analyzed.

Even if the mechanism whereby Rn initiates tumorigenesis is not completely known, literature data indicate that there are several tumor pathologies that can develop following occupational and environmental exposure. Lung cancer is certainly the most frequent disease for which the results of the literature largely confirm the etiological role of Rn and its products. However, their effect in reference to other organs is still rather elusive and sometimes contradictory. The risk depends on the concentration of Rn and its products as well as on the time spent in their presence [96], even if there is general agreement in the assumption that there is no exposure threshold value below which one can be considered protected. The probability of oncogenic phenomena, in fact, does not depend on the number of radiations that affect a single cell, but on the total number of interactions between cells and radiations, which is proportional to the exposure (International Commission on Radiation Units and Measurements, 1980.) [36].

Among other than lung cancers, we first considered brain and CNS cancer because they have been associated with ionizing radiation exposure [97,98]. The cumulative dose in brain could be a consequence to the transport of Rn and its progeny through the blood together with the fact that particles with aerodynamic diameter around 1 nm and a high extrathoracic deposition including the nasal cavity could reach the brain via the olfactory neuronal pathway [47]. Additionally, it has been proposed that macrophages might phagocyte small solid particles in the lungs that could reach the brain through the blood [99]. The weight of evidence from high-dose occupational studies suggests Rn’s potential role in brain and CNS cancer development. Nevertheless, very few studies have assessed the relationship between Rn levels and brain or CNS tumors. Most studies of mineworkers showed no relationship between occupational exposure and incidence or mortality for these cancers, while assessment of general population provided mixed results. Some papers have evaluated the impact of Rn not only in adults but also in children. In fact, CNS tumors are the second most common cancer worldwide in children aged 0–14 years, after leukemia. In addition, the United Nations Scientific Committee on the Effects of Atomic Radiation has suggested a greater risk for children compared to adults of radiation-induction of some cancer types, including CNS tumors [100]. Despite this, even in the case of children, results are inconclusive.

Similarly, risks of developing leukemia following protracted exposure to very low levels of radiation, such as those due to background radiation, are still in dispute. It has been clearly established that acute or repeated exposure to high dose ionizing radiation induces leukemia. Less is known about the effects of Rn decay alpha particles on the bone marrow. However, a recent study reported that the bronchial mucosa has an abundance of circulating lymphocytes, thus suggesting that Rn decay particles exposures could be associated with hematologic cancers originating from these cells [101]. Several studies evaluated leukemia related to occupational and environmental Rn exposure, suggesting a correlation between indoor levels and leukemia incidence at all ages [65,70,73,74,102]. With regard to childhood leukemia, the most common cancer diagnosed in children worldwide, Rn level in dwellings showed an increase in risk in numerous studies [70,71], but bias and confounding factors cannot be ruled out as possible explanations.

Environmental Rn is also relevant for skin exposure because it attaches to aerosol particles in the air, which adhere to the human skin via electrostatic attraction. Subsequently, the alpha particles from Rn decay irradiate the skin’s outer layer. Therefore, the skin receives the second-highest dose after the respiratory tract; an annual dose for the skin at 200 Bq/m^3^ is estimated to be 25 mSv [103]. The critical cells in the skin are located in the basal layer of the epidermis, and the emitted alpha particles must penetrate the outer layers to reach these cells. Eatough et al. estimated that, for nuclides attached to the actual skin surface, the dose to the basal layer is around 0.5 µSv decay-1 cm^2^ for ^218^Po, which emits a 6.0 MeV alpha particle, and 1 µSv decay-1 cm^2^ for ^214^Po, with a 7.69 MeV alpha [104]. Despite this, only scarce research on the association between skin cancer and Rn has been conducted with mixed results due to low numbers and heterogeneity of studies.

Stomach cancer has been considered because groundwater is a source of indoor air contamination due to Rn’ transfer from water to air during showers, laundry, and washing dishes. Rn dissolved in water might be riskier for stomach cells because the stomach is a storage organ and exposure might be prolonged [105]. Various models have been developed for estimating the radiation dose to different organs and tissues from ingested Rn, but there is no consensus about the doses. The NRC has proposed a conversion factor of 3.5 × 10^−9^ Sv/Bq for calculating the dose to the stomach due to Rn in drinking water [106]. The results of studies conducted to verify an association between exposure to Rn and stomach cancer are mixed, although with some prevalence for results with a positive association.

The hypothesis that Rn can cause kidney cancer is biologically plausible because the kidney filters Rn and its decay products from the blood. During this process, the radioactive alpha particles interact with renal cells directly and exert carcinogenic effects on the kidney [70]. Again, few studies have been reported about kidney cancers and Rn exposure, and these only referred to workers. Generally, no association has been found, even if a possible association cannot be definitively excluded.

Though the molecular mechanisms underlying the effects of Rn on breast cancer risk are not fully understood, Rn and its decay products have been suspected to deliver radiation doses to breast tissue [95]. In fact, based on its lipophilic properties, biokinetic models estimated the deliverable concentrations of Rn decay products detected in the breast at an annual dose of 1000 Bq/L (or 2.7 × 104 pCi/L) [99]. However, even in the case of breast cancer, few studies have been carried out about the association between Rn exposure and the development of neoplasia, and generally, increasing exposure was not associated with breast cancer risk overall.

Because of considerable experience gained by studying health effects in miners who worked in Rn-rich environments, the radioactive Rn and its progeny were identified as a cause of lung-cancer [36]. The lung is the main target organ, but it should not be considered the exclusive one. Rn enters the body principally through the respiratory system where further decay occurs, causing oxidative damage to DNA, proteins, and lipids (UNSCEAR, 2000). Rn and its progeny can settle in lungs, where they continue to undergo radioactive decay and give a radiation dose to tissues, potentially capable of irreversibly damaging the cells. Moreover, from the respiratory tree, they can pass the capillary- alveolus barrier and spread systemically to other organs. Alpha particles represent the predominant form of radiation emitted as a result of the decay of Rn (Figure 1). The most substantial alpha emitters from Rn decay are ^218^Po (6.0 MeV) and ^214^Po (7.69 MeV), and they have penetration depths of 47 µm and 70 µm, respectively [99], suggesting high levels of irradiation, particularly of the bronchial epithelium and at bifurcation sites, when inhaled into the lungs [3,107]. Inhaled decay products, largely attached to particulate matter always present in the air, settle on the walls of the respiratory system and from here irradiate, through alpha radiation, bronchial cells. Despite their limited tissue penetration capability, alpha particles can cause significant biological damage in exposed tissue due to their high relative biological effectiveness. In fact, they generate free radicals and oxidative stress, which can directly damage DNA in exposed cell nuclei. A variety of genetic lesions, including chromosomal damage, gene mutations, induction of micronuclei, and sister chromatid exchange (SCE), have been associated with the DNA damaging effects of alpha particles [108,109,110,111,112,113,114]. Rn exposure also correlates with tumor mutation burden (TMB), and subjects with a high exposure have almost twice as many mutations/Mb compared with those with low exposures. Apparently, this mutational signature is associated with defective DNA mismatch repair [115]. There is also evidence that adjacent cells may sustain damage via a “bystander effect”, phenomenon for which no irradiated cells respond to signals emitted by adjacent irradiated cells [116]. Brenner et al. suggests that the bystander effects can result in non-linear dose–response relations and inverse dose–rate effects [117]. Several identified outcomes have been attributed to the bystander effect including chromosome aberration (CAs), micronuclei formation (MN) and sister chromatid exchange (SCEs), apoptotic progression and inhibition, modification of gene and protein expression, neoplastic transformation, mutagenesis, cytokine and growth factor production, and generation of γ-H2AX species, indicative of double strand DNA breaks [112,118,119,120]. However, recently, much interest has been generated by the theory of hormesis according to which low levels of radiation exposure could have a protective effect. In the case of Rn, this theory refers to a protective effect of low-level Rn exposure against lung cancer in smokers. A possible explanation is that this low-dose exposure may eliminate the smoke-injured cells by stimulation of apoptosis and immunity [121,122,123,124].

## 5. Conclusions

Epidemiological data on Rn exposure and cancer risk confirm the lack of a clear prevalence of other than lung cancer. This is mainly due to the limited number of studies but also to the fact that they are heterogeneous with bias or several confounding factor, which makes the results less reliable. The study of Rn exposure and its health effects includes many aspects ranging from risk assessment to techniques for reduction, passing through the identification of areas with high concentrations, knowledge of the factors affecting concentrations, measurement techniques, remedial actions, and the definition of regulations. Each of these aspects requires considerable efforts and skills, and the investigation methods must always deal with the complexity of the phenomenon due to the multiplicity of sources that produce it.

The health risks associated with exposure to Rn appear to be high and require the definition of investigation plans at an international level, for further development of policies to prevent and reduce the risk. A careful evaluation and an in-depth study of the pathogenetic mechanisms that exist between exposure to Rn and the onset of pathological processes are required, in order to implement more effective primary prevention.

## Figures and Tables

**Figure 1 ijerph-18-10466-f001:**
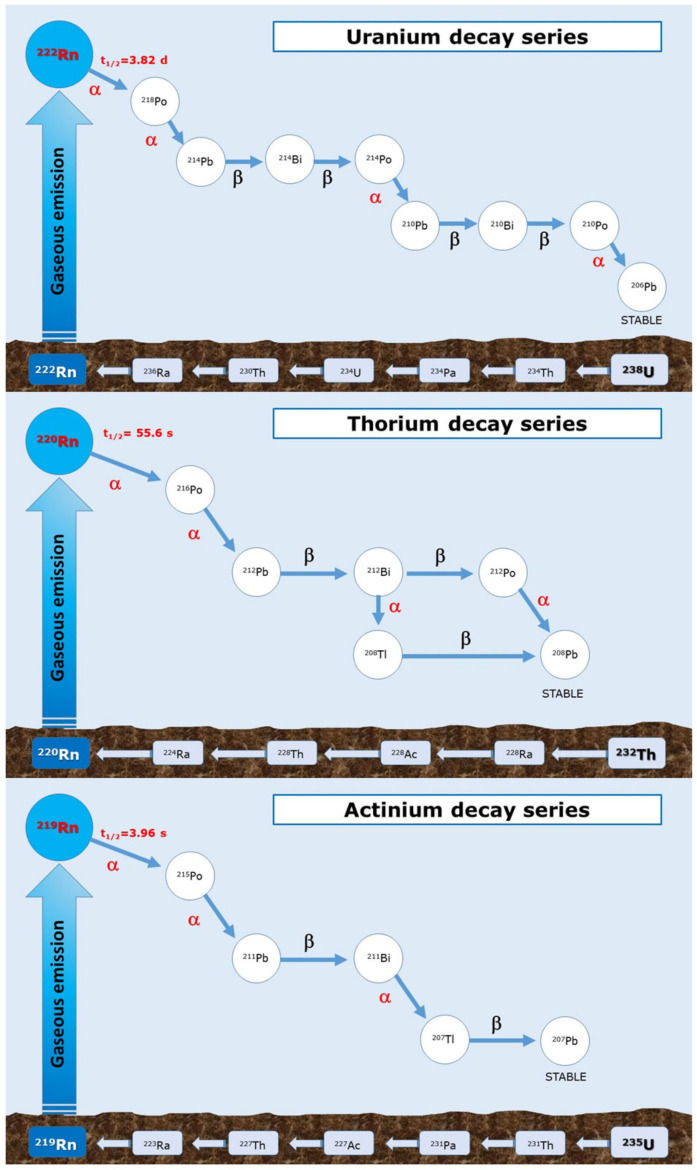
Decay series of Uranium, Thorium, and Actinium. Pa = Protactin, Ac = Actinium, Ra = Radium, Tl = Tallium.

**Table 1 ijerph-18-10466-t001:** Brain and Central Nervous System (CNS) cancer.

**a: Workers**				
**Study Design**	**Sample Size** **(*n*)**	**Radon** **Conc.**	**Results**	**Reference** **(Year)**
Cohort mortality study	1785	70.4 WLM ^1^	SMR: 1.89; 95% CI: 0.78–3.89 Expected number of deaths ^3^: *p*-value = 0.03	Tirmarche et al. (1993) [39]
Cohort mortality study	64,209	155 WLM ^1^	O/E deaths ^4^: 1.01; 95% CI: 0.95–1.07 No significant association	Darby et al. (1995) [40]
Cohort mortality study	1294	89 WLM ^1^	O/E deaths ^4^:1.21; 95% CI 1.03–1.41 No significant association	Darby et al. (1995) [41]
Cohort study	5086 (4140 exposed to radon)	36.6 WLM ^1^	No significant association	Vacquier et al. (2008) [42]
Cohort study	49,268 Ex-E 7931 NE	279.4 WLM ^2^	O/E ^4^: 1.02; 95% CI: 0.98–1.05	Kreuzer et al. (2008) [43]
Cohort study	4137	794–808 WLM	No significant association	Schubauer-Berigan et al. (2009) [44]
Cohort study	3377	17.8 WLM ^1^	SMR: 2.00; 95% CI: 1.09–3.35	Vacquier et al. (2011) [45]
**b: General Population**				
**Study Design**	**Sample Size** **(*n*)**	**Radon** **Conc.**	**Results**	**Reference** **(Year)**
Prospective study	811,961	mean ± s.d: 53.5 ± 38.0 Bq/m^3^ range: 6.3–265.7	HR: 0.98 per 100 Bq/m^3^; 95% CI: 0.83–1.15 No clear associations	Turner et al. (2012) [46]
Cohort study	57,053	40.5 Bq/m^3^	IRR: 1.96; 95% CI: 1.07–3.58	Bräuner et al. (2013) [47]
Ecological study	251	GM: 100–200 Bq/m^3^	Spearman’s Rho: 0.286 (males; *p*-value: <0.001;) 0.509 (females; *p*-value: <0.001)	Ruano-Ravina et al. (2017) [48]
Ecological study	13	153.9 Bq/m^3^	RR: 1.28 Statistical association	López-Abente et al. (2018) [49]
Ecological study	New Jersey: 14,662; Iowa: 8429; Wisconsin: 8023; Pennsylvania: 22,940; Minnesota: 5338	4.6–8.6 pCi/L	Negative association: *p*-value <0.0001	Monastero et al. (2020) [50]
**c: Pediatric Population**				
**Study design**	**Sample Size** **(*n*)**	**Radon** **Conc.**	**Results**	**Reference** **(Year)**
Ecological study	Total death: 2706 Brain and CNS disease: 454	0–10,692 pCi/l ^1^	Medium exposure RR: 1.28; 95% CI: 1.00–1.62 High exposure RR: 1.18; 95% CI: 0.90–1.54	Collman et al. (1991) [51]
Case-control study	82 L; 82 ST; 209 Controls	mean: 27 Bq/m^3^ range: 10–584 Bq/m^3^	Solid Tumor OR: 2.61; 95% CI: 0.96–7.13	Kaletsch et al. (1999) [52]
Case-control study	Cases: 2400 Controls: 6697	mean: 48 Bq/m^3^ range: 4–254 Bq/m^3^	No significant association	Raaschou-Nielsen et al. (2008) [53]
Cohort study	Childhood cancer cases: 997	median: 77.7 Bq/m^3^; 90th: 139.9 Bq/m^3^	All cancers HR: 0.93; 95% CI: 0.74–1.16 CNS tumors HR: 1.05; 95% CI: 0.68–1.61	Hauri D et al. (2013) [54]
Case-control study	Cases: 27,447 Controls: 36,793	mean: 22 Bq/m^3^	ERR: 3%; 95% CI: 4–11; *p*-value: 0.35	Kendall et al. (2013) [55]
Cohorts	Total: 712,674 Cancer cases: 864	mean: 91 Bq/m^3^ median: 74 Bq/m^3^	<50 Bq/m^3^ HR: 1.00 (Ref.) 50–100 Bq/m^3^ HR: 0.88; CI: (0.68–1.14) >100 Bq/m^3^ HR: 1.15; CI: (0.87–1.50) No significant association	Del Risco Kollerud et al. (2014) [56]
Ecological study	5471 cases of CNST	^1^ 41.0 Bq/m^3^	IRR: 1.07; CI: 0,95–1.20 per 100 Bq/m^3^ No significant association	Berlivet J et al. (2020) [57]

a: ^1^ Average cumulative exposure; ^2^ accumulated exposure; ^3^ malignant brain tumor and malignant tumors of other parts of the nervous system; ^4^ for all other than lung cancer combined; WLM: Working Level Month; Ex-E: ex esposed; NE: never exposed; O/E: observed/expected cases; 95% CI: 95% confidence interval; SMR: Standardized Mortality Ratio. b: IRR: incidence rate-ratios; RR: Relative Risk; GM: geometric mean; 95% CI: 95% confidence interval; HR: Hazard Ratio. c: ^1^ drinking water; L: leukemias; ST: solid tumors; IRR: incidence rate-ratios; RR: Relative Risk; 95% CI: 95% confidence interval; HR: Hazard Ratio; OR: odds ratio; ERR: excess relative risk.

**Table 2 ijerph-18-10466-t002:** Radon exposure and leukemia.

**a: Workers**				
**Study** **Design**	**Sample Size** **(*n*)**	**Radon** **Conc.**	**Results**	**Reference** **(Year)**
Cohort mortality study	4320	196.8 WLM ^1^	O/E deaths: 1.11; 95% CI: 0.98–1.24 No significant association	Tomàsek et al. (1993) [61]
Cohort study	64,209	155 WLM ^1^	O/E: 1.93; 95% CI: 1.19–2.95 No significant association	Darby et al. (1995) [40]
Retrospective case–cohort study	23,043	mean ± sd: 64.1 ± 98 WLM	All leukemia RR: 1.75; 95% CI: 1.10–2.78; *p*-value = 0.014 CLL RR: n.s.	Rericha et al. (2006) [62]
Cohort study	58,987	279.4 WLM	No significant association O/E: 0.89; 95% CI: 0.74–1.06	Kreuzer et al. (2008) [43]
Cohort study	17,660	100.2 WLM	All leukemia SMR: 0.69; 95% CI: 0.48–0.97; *p*-value = 0.031 SIR: 0.79; 95% CI: 0.59–1.03; *p*-value = 0.088	Zablotska et al. (2014) [63]
Cohort study	16,434	53 WLM	All leukemias: SIR: 1.51; 95% CI: 1.08–2.07 Lymphatic and hematopoietic cancers combined SIR: 1.31; 95% CI: 1.05–1.61	Kelly-Reif et al. (2019) [64]
**b: General and Pediatric Population**				
**Study** **Design**	**Sample Size** **(*n*)**	**Radon** **Conc.**	**Results**	**Reference** **(Year)**
Ecological study	45	<120 Bq/m^3^	Lymphocytic leukemia r = 0.40; *p*-value < 0.005, ρ = 0.24; *p*-value < 0.1 Myeloid leukemia r = 0.43; *p*-value < 0.005, ρ = 0.22 *p*-value < 0 1	Eatough et al. (1993) [65]
Ecological study	Area ≥ 100 Bq/m^3^ Cases: 35 Area < 100 Bq/m^3^ Cases: 73	Area ≥ 100 Bq/m^3^ (Mean: 183 Bq/m^3^) Area < 100 Bq/m^3^ (Mean: 57 Bq/m^3^)	Area ≥ 100 Bq/m^3^: Incidence = 106.7 per million child years Area < 100 Bq/m^3^: Incidence = 121.7 per million child years No significant difference between Area ≥ 100 Bq/m^3^ and Area < 100 Bq/m^3^ (*p*-value = 0.29).	Thorne et al. (1996) [66]
Case-control study	Cases: 173 Controls: 254	^1^ Cases: 56.0 Bq/m^3^ ^1^ Controls: 49.8 Bq/m^3^	37–100 Bq/m^3^ adjusted OR: 1.2; 95% CI: 0.7–1.8 >100 Bq/m^3^ adjusted OR: 1.1; 95% CI 0.6–2.0 No significant difference	Steinbuch et al. (1999) [67]
Case-control study	Cases: 505 Controls: 443	^1^ Cases 65.4 Bq/m^3^ ^1^ Controls 79.1 Bq/m^3^	Rn concentration < 37 Bq/m^3^ RR: 1; (Reference) Rn concentration 37–73 Bq/m^3^ RR: 1.22; 95% CI: 0.8–1.9 Rn concentration 74–147 Bq/m^3^ RR: 0.82; 95% CI: 0.8–1.9 Rn concentration ≥ 148 Bq/m^3^ RR: 1.02; 95% CI: 0.5–2.0	Lubin et al. (1998) [68]
Case-control study	Cases: 82 Controls: 209	Median: 27 Bq/m^3^ Range: 10–584 Bq/m^3^	OR: 1.30; 95% CI: 0.32–5.33	Kaletsch et al. (1999) [52]
Case-control study	Cases: 3838 cases (1461 ALL) Controls 7629	^1^ 24.0 Bq/m^3^	OR: 0.80; 95% CI: 0.64–0.99	UKCCS [69] (2002)
Ecological study	Data not provided	UK: 20 Bq/m^3^; Cornwall: 110 Bq/m^3^; World: 50 Bq/m^3^	Country data alone r = 0.65; *p*-value < 0.02; Regional data r = 0.62; *p*-value <0.02	Henshaw et al. (1990) [70]
Ecological study	Leukemias and lymphomas: 4851	median: 21 Bq/m^3^,	RR: 1.06; 95% CI: 0.99–1.12	Gilman et al. (1998) [71]
Ecological correlation study	53,146	High risk area: 50,000 Bq/m^3^; Normal risk area: 10,000–50,000 Bq/m^3^; Low risk area: <10,000 Bq/m^3^;	ALL^2^: RR (normal risk area): 4.64 95% CI: 1.29–28.26 RR (high risk area): 5.67 95% CI: 1.06–42.27	Kohli et al. (2000) [72]
Cohort study	Childhood cancer cases: 997	77.7 ^1^ Bq/m^3^, 90th: 139.9 Bq/m^3^	All leukemias AHR: 0.90; 95% CI: 0.56–1.43 Acute lymphoblastic leukemia AHR: 0.90; 95% CI: 0.56- 1.43	Hauri D et al. (2013) [54]

a: ^1^ Average cumulative exposure; ^2^ lifetime Rn exposure. WLM: Working Level Month; O/E: observed/expected cases; 95% CI: 95% confidence interval; RR: Relative Risk; CLL: chronic lymphocytic leukemia; ALL = acute lymphatic leukemia; AML: acute myeloid leukemia; n.s.: not significant. b: ^1^ Arithmetic mean of time-weighted radon concentrations; ^2^ ALL: acute lymphatic leukemia; RR: Relative Risk; 95% CI: 95% confidence interval; OR: odds ratio; r: correlation coefficient; AHR: Adjusted Hazard Ratio.

**Table 3 ijerph-18-10466-t003:** Skin cancer: General Population.

Study Design	Sample Size (*n*)	Radon Conc.	Results	Reference (Year)
Ecological study	28,989	^1^ 40 Bq/m^3^ ^1^ ≥230 Bq/m^3^	Non-melanoma skin cancers, showed a significant increase in incidence in the high-radon postcode sectors (≥100 Bq/m^3^) compared with the low-radon sectors (<60 Bq/m^3^) and this effect was observed for both sexes.	Etherington et al. (1996) [78]
Prospective study	811,961	mean ± sd: 53.5 ± 38.0 Bq/m^3^ range: 6.3–265.7 Bq/m^3^	HR = 0.98; 95% CI: 0.97–1.00, per each 100 Bq/m^3^ no significant association	Turner et al. (2012) [46]
Ecological study	18,350	mean ± sd: 98.1 ± 73.1 Bq/m^3^	Malignant melanoma Rn concentration ≥ 230 Bq/m^3^ RR: 0.85; 95% CI: 0.65–1.11 Basal cell carcinoma Rn concentration ≥ 230 Bq/m^3^ RR: 0.81; 95% CI: 0.66–1.00 Squamous cell carcinoma Rn concentration ≥ 230 Bq/m^3^ RR: 1.76; 95% CI: 1.46–2.11	Wheeler et al. (2012) [79]
Ecological study	206,454	range: 0–≥100 Bq/m^3^	0.18^2^ registrations per 100,000 population per year 95% CI: 0.04–0.32 *p*-value = 0.011	Wheeler et al. (2013) [80]
Prospectic cohort	57,053	median: 38.3 Bq/m^3^	BCC Adjusted IRR: 1.14; 95% CI: 1.03–1.27 SCC Adjusted IRR: 0.90; 95% CI: 0.70- 1.37 MM Adjusted IRR: 1.08; 95% CI: 0.77–1.50	Bräuner et al. (2015) [81]
Ecological study	2294	Cutpoint set: 50 Bq/m^3^	Risk of non-pulmonary cancer HR: 1.2; 95% CI: 0.9–1.6 Connective tissue and others of the skin HR: 1.5; 95% CI: 0.6–3.8 (except melanoma)	Barbosa-Lorenzo et al. (2016) [82]
Mortality cohorts	Tot: 5,249,462 skin cancer deaths: 2989	mean ± sd: 91.8 ± 47.8 Bq/m^3^	MM HR^3^: 1.41; 95% CI: 1.09–1.80 at 30 years HR^3^: 1.05; 0.94–1.18 at 75 years Adjusted HR: 1.16; 95% CI: 1.04–1.29 at 60 years	Vienneau et al. (2017) [83]

^1^ Average radon levels for postcode sectors; ^2^ adjusted coefficient per 1 Bq/m^3^ increase; ^3^ per 100 Bq/m^3^; 95% CI: 95% confidence interval; sd: standard deviation; IRR: incidence rate-ratios; HR: Hazard Ratio. BCC: basal cell carcinoma; SCC: squamous cell carcinoma; MM: malignant melanoma.

**Table 4 ijerph-18-10466-t004:** Stomach cancer: Workers and General Population.

Study Design	Sample Size (*n*)	Radon Conc.	Results	Reference (Year)
Cohort study	64,209	155 WLM	O/E: 1.33; 95% CI: 1.169–1.52 Mortality: no significant association	Darby et al. (1995) [40]
Cohort study	1415	89 WLM	O/E: 1.45; 95% CI: 1.04–1.98 Mortality: no significant association	Darby et al. (1995) [41]
Cohort study	28,546	21 WLM	Incidence: Overall excess ^1^ RR: −0.12; 95% CI: −0.59–0.35 Mortality: overall excess ^1^ RR: −0.082; 95% CI: −0.61–0.45 No significant association	Navaranjan et al. (2016) [86]
Ecological study	5218	Groundwater: 100 Bq/L Indoor air: 100 Bq/m^3^	Groundwater Rn exposure and stomach cancer OR: 1.24; 95% CI 1.03–1.49	Messier et al. (2017) [87]
Ecological study	56,385	IQ range: 53–184 Bq/m^3^	Statistical association in women RR: 1.17; 95% CI 1.02–1.32	López-Abente et al. (2018) [49]
Cohort study	16,434	53 WLM	SMR: 1.27; 95% CI: 1.02–1.51 SIR: 1.37; 95% CI: 1.11–1.63	Kelly-Reif et al. (2019) [64]

^1^ Per 100 WLM; WML: Working Level Month; 95% CI: 95% confidence interval; O/E: observed/expected cases; SMR: Standardized Mortality Ratio; SIR: Standardized Incidence Ratio; RR: relative risk; IQ: interquartile range; OR: Odd ratio.

**Table 5 ijerph-18-10466-t005:** Kidney cancer: Workers.

Study Design	Sample Size (*n*)	Radon Conc.	Results	Reference (Year)
Cohort study	5086	36.6 WLM	A significant excess of kidney cancer deaths was observed (*n* = 20; SMR = 2.0; 95% CI: 1.22–3.09), which was not associated with cumulative Rn exposure	Vacquier et al. (2008) [42]
Cohort study	28,546	21 WLM	No association for kidney cancer with increasing cumulative Rn exposure	Navaranjan et al. (2016) [86]
Cohort study	French cohort: 3377; German cohort: 58,986	median: 4.7 WLM range: 0–128.4 WLM median: 18.4 WLM range: 0–3224.5 WLM	French cohort SMR: 1.49; 95% CI: 0.73–2.67 German cohort SMR = 0.91; 95% CI: 0.77–1.06 No significant association	Drubay et al. (2014) [90]

WML: average cumulative exposures; 95% CI: 95% confidence interval; SMR: Standardized Mortality Ratio.

**Table 6 ijerph-18-10466-t006:** Breast cancer: Workers and general population.

Study Design	Sample Size (*n*)	Radon Conc.	Results	Reference (Year)
Cohort study	School employees: 520	Classrooms: <2 pCi/L Utility tunnels: 29–33 pCi/L	SMR: 5; 95% CI: 1.03–14.6	Neuberger et al. (1997) [92]
Observational cohort study	74,806	^1^ water: 1.3–9 Bq/L	HR: 1.59; 95%CI: 1.10–2.31	Kristbjornsdottir et al. (2012) [93]
Prospective cohort study	811,961	mean ± sd: 53.5 ± 38.0 Bq/m^3^ range: 6.3–265.7 Bq/m^3^	HR: 0.91; 95% CI: 0.82–1.01	Turner et al. (2012) [46]
Prospective cohort study	112,639 female	^2^ P20: <27.0 Bq/m^32^ P40: ≥27.0–37.7 Bq/m^3^ ^2^ P60: ≥37.7–50.1 Bq/m^3^ ^2^ P80: ≥50.1–74.9 Bq/m^3^ ^2^ P100: ≥74.9 Bq/m^3^	For women in the highest quintile of exposure (≥74.9 Bq/m^3^) HR = 1.38; 95% CI: 0.97–1.96 (ER-/PR-) No association was observed for ER+/PR+ breast cancer	VoPham et al. (2017) [94]

^1^ Geothermal hot water from drilled wells; ^2^ P20: 1st quintile, P40: 2nd quintile, P60: 3rd quintile, P80: 4th quintile, and P100: 5th quintile; 95% CI: 95% confidence interval; SMR: Standardized Mortality Ratio; HR: Hazard Ratio; ER-/PR-: Estrogen Receptor negative/Progesterone receptor negative; ER+/PR+: Estrogen Receptor negative/Progesterone receptor negative.

## Data Availability

Not applicable.

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
