# Peer review of "Environmental/Occupational Exposure to Radon and Non-Pulmonary Neoplasm Risk: A Review of Epidemiologic Evidence"

_ijerph, 2021, doi:10.3390/ijerph181910466_

Round 1

Reviewer 1 Report

Thank you for the opportunity to review the manuscript titled “Environmental/occupational exposure to Radon and non-pulmonary neoplasm risk: a review of epidemiologic evidence”. The manuscript is well written and reads well. The relationship of radon with lung cancer is well established but little is known about its association with other forms of cancers.  Radon exposure has shown some increased incidence of chromosomal aberrations in miners in previous studies however the causal relation with non-pulmonary cancers is thus is an important one. The works summarizes and analyses the current available studies well. I appreciate the breakdown based on different cancers. Even though the many data may not show conclusive evidence but in absence of a statistical analysis I would caution against wordings as in the abstract like “Overall, the available evidence does not support a conclusion that a causal association has been established between Rn exposure and the risk of other non-pulmonary neoplasms” as it conclusion is not supported by statistical analysis.  

Author Response

We thank the reviewer for this correct observation. Therefore, the following sentence has been added at the end of the Abstract:

“To confirm this result, a statistical analysis should be necessary, even if it is now not applicable for the few studies available.”

Reviewer 2 Report

This epidemiological review by Mozzoni and coauthors addresses the environmental/occupational exposure to Rn and non-pulmonary neoplasm risk, which makes it interesting and complete in its field because it considers both different populations and levels of exposure. 

I have minor comments:

  1. lanes 95 and 99: [96/29/Euratom; [21], [2013/59/Euratom;[22]. I think there is a mistake in the format of these references.
  2. lane 142. what mean IARC? To definy this acronym.
  3. lane 191. γ rays should be γ-rays.

Author Response

We thank the reviewer for these observations. Particularly,

  1. Line 95 and 99 - [96/29/Euratom and [2013/59/Euratom have been replaced by [96/29/Euratom] and [2013/59/Euratom], respectively.
  2. Line 142: “International Agency for Research on Cancer” has been added.
  3. Line 191: γ rays has been replaced by γ-rays

Reviewer 3 Report

Dear authors, your work is very nice and interesting, it is quite well-written. It addresses the different types of cancer caused by Radon exposure.

I recommend it for publication, but you should try to follow these minor recommendations:

1.- Please, when you have to indicate several references in a row, you must use only one clasp to open and other clasp to close, and the numbers of the references separated with coma, for example, references of Line 69: [11, 12, 13, 14]

2.- The work is very-very well structured, the section of Results is very clear, and it helps the reader to understand the work. All the work and its structure are presented in a quite coherent and organized manner. But you should include a paragraph at the end of the Introduction Section in order to present the different parts of the work, (including a very little resume of the well-structured Results Section) in order to present the reader what he/she is going to find through the paper.

Author Response

We thank the reviewer for his observations.

  1. All references have been checked and corrected as indicated by the reviewer.

  1. We agree with the reviewer that a brief introduction to the presentation of the results would facilitate reading. Therefore, the following sentence has been added:

“Particularly, leukemia and tumors related to brain and central nervous system (CNS), skin, stomach, kidney and breast have been investigated focusing on workers, general population and paediatric population.”